# Atomic Insights into Ti Doping on the Stability Enhancement of Truncated Octahedron LiMn_2_O_4_ Nanoparticles

**DOI:** 10.3390/nano11020508

**Published:** 2021-02-17

**Authors:** Wangqiong Xu, Hongkai Li, Yonghui Zheng, Weibin Lei, Zhenguo Wang, Yan Cheng, Ruijuan Qi, Hui Peng, Hechun Lin, Fangyu Yue, Rong Huang

**Affiliations:** 1Key Laboratory of Polar Materials and Devices (MOE) and Department of Electronics, East China Normal University, Shanghai 200062, China; wangqiongx@126.com (W.X.); hongkaizzz@foxmail.com (H.L.); yhzheng@phy.ecnu.edu.cn (Y.Z.); weibinleilove@126.com (W.L.); wzgcoinmail@126.com (Z.W.); ycheng@ee.ecnu.edu.cn (Y.C.); hpeng@ee.ecnu.edu.cn (H.P.); hclin@ee.ecnu.edu.cn (H.L.); fyyue@ee.ecnu.edu.cn (F.Y.); 2Collaborative Innovation Center of Extreme Optics, Shanxi University, Taiyuan 030006, Shanxi, China

**Keywords:** truncated octahedral LiMn_2_O_4_, Ti doping, crystal planes, cathode materials, Li-ion batteries

## Abstract

Ti-doped truncated octahedron LiTi*_x_*Mn_2-*x*_O_4_ nanocomposites were synthesized through a facile hydrothermal treatment and calcination process. By using spherical aberration-corrected scanning transmission electron microscopy (Cs-STEM), the effects of Ti-doping on the structure evolution and stability enhancement of LiMn_2_O_4_ are revealed. It is found that truncated octahedrons are easily formed in Ti doping LiMn_2_O_4_ material. Structural characterizations reveal that most of the Ti^4+^ ions are composed into the spinel to form a more stable spinel LiTi*_x_*Mn_2−*x*_O_4_ phase framework in bulk. However, a portion of Ti^4+^ ions occupy 8a sites around the {001} plane surface to form a new TiMn_2_O_4_-like structure. The combination of LiTi*_x_*Mn_2−*x*_O_4_ frameworks in bulk and the TiMn_2_O_4_-like structure at the surface may enhance the stability of the spinel LiMn_2_O_4_. Our findings demonstrate the critical role of Ti doping in the surface chemical and structural evolution of LiMn_2_O_4_ and may guide the design principle for viable electrode materials.

## 1. Introduction

Rechargeable lithium-ion batteries (LIBs) have been regarded as promising energy storage and conversion devices for wearable mobile devices, electric vehicles (EVs), hybrid electric vehicles (HEVs), and stationary energy storage wells [1,2,3]. Among the various lithium-ion battery cathode materials, spinel LiMn_2_O_4_ is believed to hold huge potential for fulfilling the field-use requirements because of its good thermal stability, low cost, environmental friendliness, and three-dimensional channel structure [4,5,6]. Nevertheless, the practical applications of LiMn_2_O_4_ cathodes are restricted by the capacity fading during charge–discharge cycles, especially at elevated temperatures (≥ 55 °C), which can be ascribed to the Mn dissolution and Jahn–Teller distortion [7,8].

In order to tackle these challenges, efforts have been paid to stabilize the structure of LiMn_2_O_4_. By doping with monovalent (e.g., Li^+^ [9]), divalent (e.g., Mg^2+^ [10] and Ni^2+^ [11]) or trivalent (e.g., Al^3+^ [12], Co^3+^ [13] and Fe^3+^ [14]) metal ions, the average manganese ion valence is slightly increased, and therefore the Jahn–Teller effect is suppressed and a promoted cycling performance is obtained in the 4 V region. However, when LiMn_2_O_4_ works in the 2.0–4.8 V, inactive Mn^4+^ ions in the 4 V regions are further reduced to Mn^3+^ ions, and the cycle performance of low-valent ions doped materials is not that satisfactory. For example, Lee et al. [15] found that the LiAl_0_._1_Mn_1_._9_O_4_ achieved capacity retention of 70% after 50 cycles in the 2.0–4.3 V range. When cycled between 2.0 and 5.0 V, the LiNi_0_._5_Mn_1_._5_O_4_ shows a capacity retention value of about 65% [16]. Since the bond energy of Ti-O (662 kJ mol^−1^) is higher than that of Mn-O (402 kJ mol^−1^), the Mn^4+^ in the lattice of LiMn_2_O_4_ could be partly replaced by Ti^4+^ to form a more stable spinel framework, i.e., [Mn_2−*x*_Ti*_x_*]O_4_, therefore enhancing the stability of the spinel LiMn_2_O_4_. Recently, He et al. [17] reported that 72% capacity retention was achieved with the LiTi_0_._5_Mn_1_._5_O_4_ electrode after 150 cycles performed between 2.0 and 4.8 V. By using an in situ X-ray diffraction technique, Wang et al. [18] found that Ti^4+^ ions can also suppress the Jahn–Teller distortion and stabilize the spinel structure during the charging/discharging process. Moreover, Ti substitution improves the structural stability of spinel cathode material as reported at large [19,20,21]. Although these findings are important and intriguing, a deep understanding on how Ti doping contributes to the stability enhancement of LiMn_2_O_4_ is still lacking.

To date, various experimental and computational results show that the structural stability of LiMn_2_O_4_ is strongly related to its surface structure [22,23,24,25]. Karim et al. [26] ascribed the improved stability of LiMn_2_O_4_ to the creation of a partial inverse spinel arrangement in the (111) surface. A further example by Ouyang et al. [27] showed that covering the LiMn_2_O_4_ (001) surface with Al_2_O_3_ changed the oxidation state of surface Mn atoms from +3 to +4, which is beneficial for the improvement in LiMn_2_O_4_ stability. Nevertheless, few studies have been undertaken to reveal the surface structure and chemical evolution of Ti-doped LiMn_2_O_4_ at atomic levels.

In this work, Ti-doped truncated octahedron LiMn_2_O_4_ samples are synthesized through a facile hydrothermal treatment and calcination process. To reveal the underlying mechanism of Ti-doping on the structure evolution and stability enhancement of LiMn_2_O_4_, morphology and phase characterization are performed by powder X-ray diffraction (XRD), scanning electron microscope (SEM), and Raman spectroscopy. X-ray photoelectron spectroscopy (XPS) further reveals that Ti ions are in a tetravalent oxidation state; after Ti ion doping, the percentage of Mn^4+^ in LiTi_0_._5_Mn_1_._5_O_4_ reduced, suggesting the successful replacement of Mn^4+^ by Ti^4+^. The surface evolution of LiTi*_x_*Mn_2−*x*_O_4_ (001) planes was investigated using the spherical aberration-corrected scanning transmission electron microscopes (Cs-STEM) technique. It is found that there is a more stable spinel LiTi*_x_*Mn_2−*x*_O_4_ formed in bulk, as well as at the {111} and {110} planes. In addition, for the first time, a TiMn_2_O_4_-like structure formed at {001} surface is observed by the Cs-STEM technique, which can reduce the surface energy of {001} planes and accelerate the growth rate of {001} planes. In addition, the TiMn_2_O_4_-like structure at {001} surface might improve the stability of LiMn_2_O_4_. According to the electron energy-loss spectroscopy (EELS) analysis, the appearance of the TiMn_2_O_4_-like phase is associated with the enrichment of Ti^4+^. This work provides a comprehensive understanding of the influence of Ti doping on the evolutions of morphology, surface structure, and electronic structure of LiMn_2_O_4_ cathodes, which will benefit the further optimization of the electrochemical performance.

## 2. Materials and Methods

### 2.1. Sample Preparation

The LiTi*_x_*Mn_2−*x*_O_4_ (0 ≤ x ≤ 0.5) samples were synthesized by hydrothermal treatment and a calcination process [28,29], as depicted in Figure 1. First, to get Mn_3_O_4_ nanoparticles with better reaction activity and smaller particle size, commercially purchased Mn_3_O_4_ powders (1.0 g) were dispersed into NaOH aqueous solution (30 mL, 5 mol dm^−3^) and magnetically stirred for 1 h. Afterward, the dispersion was transferred to a Teflon-lined stainless-steel autoclave (50 mL) and heated at 205 °C for 4 d in an oven. The final precipitated products were washed repeatedly with deionized water. The obtained Mn_3_O_4_ precursor was subsequently dried at 70 °C for 12 h in air. Then, the as-prepared Mn_3_O_4_ precursor, LiNO_3_, LiCl·H_2_O, and TiO_2_ (rutile) were ground in a mortar for 30 min and burned in the air at 500 °C for 3 h. The obtained Ti-doped LiMn_2_O_4_ precursors were washed repeatedly in deionized water to remove chlorion and nitrate impurities. Finally, the obtained Ti-doped LiMn_2_O_4_ precursor was calcined in air at 700 °C for 6 h. The final products were obtained after cooling to room temperature. 

### 2.2. Sample Characterization

The crystal structures were characterized by X-ray diffraction (XRD, D8, Bruker, Germany) with Cu Kα radiation; the data were collected between 10 and 80 degrees at an increment of 0.02 degrees. The size and morphology of the samples were observed by scanning electron microscope (SEM, S-4800, Hitachi, Japan). The crystal quality and defects were characterized by Raman spectra using a micro-Raman spectrometer (Jobin Yvon LabRAM HR 800UV, Longjumeau, France) with a 532 nm laser source. EDS mapping was performed with an Oxford Inca EDS detector on the JEOL 2100F, operated in the dark field scanning transmission electron microscopy (STEM, JEOL) mode. X-ray photoelectron spectroscopy (XPS, Thermo Fischer, ESCALAB 250Xi, Walham, MA, USA) measurements were performed to investigate the valence states of the materials, using the value of 284.8 eV as the C 1s peak reference. High-angle annular dark-field scanning transmission electron microscopy (HAADF-STEM) imaging and electron energy-loss spectroscopy (EELS) were performed with a spherical aberration-corrected (Cs-corrected) scanning transmission electron microscopy (STEM) operated at 300 kV (JEM-ARM300F, JEOL). 

## 3. Results and Discussions

The phase and crystal structure of the samples are examined by XRD, as shown in Figure 2a. The diffraction peaks of all samples can be indexed to the standard pattern of spinel LiMn_2_O_4_ (JCPDS card No.35-0782; space group Fd-3m (No. 227)) without any impurity phases. More importantly, the relative peak intensities reflect the dominant surface orientations of each sample. Compared to the octahedron, the peaks on the (400), (440), and (311) lattice planes (Appendix A) are more obvious in truncated octahedron samples after normalizing peaks to the dominant (111) octahedral orientation. With the increase in Ti content, the diffraction peaks shift toward lower angles, suggesting the increase in lattice parameters. The detailed lattice parameters of the LiTi*_x_*Mn_2−*x*_O_4_ (*x* = 0, 0.1, 0.2, 0.3, 0.4, 0.5) samples were calculated and are listed in Appendix A. Since the atom radius of Ti^4+^ (0.061 Å) is larger than that of Mn^4+^ (0.053 Å) [30], the enlargement of the lattice constant indicates the substitution of Mn^4+^ by Ti^4+^ in the lattice, in line with previous reports [31].

Furthermore, the microstructure vibration of LiTi*_x_*Mn_2−*x*_O_4_ with different Ti doping content (Figure 2b) is investigated by Raman spectroscopy. The medium peak at about 480 cm^−1^ has F_2g_^(2)^ symmetry, while the weak bands observed at 400 and 370 cm^−1^ have the E_g_ and F_2g_^(3)^ symmetry, respectively [32,33]. The weak peak at 370 cm^−1^ is related to the Li-O symmetric vibration, i.e., connecting to the tetrahedral cation movements (F_2g_^(3)^) [34]. A very weak band at 285 cm^−1^ might be associated with the translation mode of lattice vibration [35]. A strong Raman peak at ~640 (±5) cm^−1^ could be assigned the symmetric Mn–O stretching vibration of [MnO_6_] octahedron (A_1g_ mode). Moreover, a blue shift below *x* = 0.2 and a redshift above *x* = 0.2 are observed (Appendix A), which further confirms the substitution of Ti atoms.

As shown in Figure 3a-f, the surface morphology and particle size of LiTi_x_Mn_2-x_O_4_ (x = 0, 0.1, 0.2, 0.3, 0.4, 0.5) were studied by SEM. The pristine LiMn_2_O_4_ (Figure 3a) is the prototype octahedral shape, which is bounded by eight {111} planes. It is reported that the truncated octahedral structure is beneficial for improving the high-rate capability and prolonging the cycle stability of LIBs, as the {111} planes can mitigate Mn dissolution while the truncated {110} and {001} planes facilitate Li^+^ diffusion [8]. Though several strategies have been proposed to obtain truncated octahedral structures [5,36,37,38], in this report we find that Ti doping is beneficial to synthesize a truncated octahedral shape. With the increase in Ti concentration, the growth rate is increased in the (001) plane (red rectangle), reduced in the (111) plane (blue lines), and remains the same in the (110) plane (green lines), implying that the Ti doping can reduce the surface energy of the (001) planes. In addition, the particle size is also increased with the doping of the Ti element (Appendix A), which may result from the substitution of Ti with Mn element.

To further examine the valence states of elements in the mixed-valence compounds, XPS was performed for LiMn_2_O_4_ and LiTi_0_._5_Mn_1_._5_O_4_, respectively. Figure 4a shows that the peaks of Ti 2p_3/2_ and Ti 2p_1/2_ in LiTi_0_._5_Mn_1_._5_O_4_ are located at 458.2 and 463.8 eV, respectively, with 5.6 eV spin-orbit components, indicating that the Ti ions are in the tetravalent oxidation state [31,39]. As for the Mn 2p XPS spectra, two main peaks corresponding to the spin-orbit splitting of Mn 2p_3/2_ and Mn 2p_1/2_ are observed in LiMn_2_O_4_ and LiTi_0_._5_Mn_1_._5_O_4_ (Figure 4b) [40]. Since the full width at half-maximum (FWHM) of the Mn 2p_3/2_ peaks are both larger than 3.5 eV, the oxidation states of Mn are expected to be between +3 and +4 valence. Furthermore, curve-fitting was conducted on the Mn 2p_3/2_ spectra (Figure 4c,d, see the fitting parameters in the supporting information in Appendix A) to evaluate the percentage of Mn^3+^ and Mn^4+^ ions [41,42]. The results show that the concentration of Mn^4+^ reduces from 47.28% to 42.13%, while the concentration of Mn^3+^ increases from 52.72% to 57.86% due to the substitution of Ti^4+^ ions (*x* = 0.5), as observed in Figure 4a.

To reveal the underlying mechanism of Ti-doping on the structure evolution and stability enhancement of LiMn_2_O_4_, samples with different Ti doping concentrations were systematically investigated using the Cs-STEM technique [43]. Figure 5a verifies the octahedron characteristic of LiMn_2_O_4_ composed of {111} facets, and Figure 5b is the enlarged HAADF image, taken along the [110] direction around the (111) surface. Since the contrast of the HAADF-STEM image is roughly proportional to the square of the atomic number Z [44], the Li (Z = 3) and O (Z = 8) are invisible, while the Mn (Z = 25) could be detected. The Mn diamond configuration was clearly observed (Figure 5b), in line with the previously reported [45], showing a homogeneous microstructure from the bulk to the surface. 

As for the LiTi_0_._5_Mn_1_._5_O_4_, truncated octahedrons composed of {111} facets, {110} and {001} facets were observed. A uniform distribution of the Mn, O, and Ti elements is also shown in Appendix A. Similar to Figure 5b, the spinel crystal structure in LiTi_0_._5_Mn_1_._5_O_4_ is also stable from the bulk to the surface in (111) planes according to the HAADF image (Figure 5d) taken along [110] orientation. This homogenous situation also happened in the (110) plane, as depicted in Figure 5e. However, a phase transition from the bulk to the surface appears progressively in (001) planes, as indicated by the cyan line. Though the atomic configuration in the bulk region (red rectangle) is similar to Figure 5d–e, the surface region (purple rectangle) is quite different. The contrast of the atoms at Li tetrahedral sites becomes brighter and visible, which can be attributed to the substitution of heavy Ti or Mn (TM) ions [46]. This is also further confirmed by the line profiles shown in Figure 5i, in which the spacing in the surface area (d = 8.69 Å) is larger than the bulk area (d = 8.20 Å).

We inspected the crystal structure of LiMn_2_O_4_, TiMn_2_O_4_, and Mn_3_O_4_ along the [110] direction, as shown in Figure 6a–c. Though the atomic arrangement is similar, the long diagonals (*n*) for TiMn_2_O_4_ (*n* = 8.679 Å) is significantly higher than that of LiMn_2_O_4_ (*n* = 8.245 Å) and Mn_3_O_4_ (*n* = 8.15 Å). Thus, the new phase formed at (001) surface is expected to be TiMn_2_O_4_, which can help to combat the impedance growth [47] and promote the electrochemical performance of high-voltage spinel LiNi_0_._5_Mn_1_._5_O_4_. In short, the majority of the Ti atoms could replace Mn element in the bulk area and form a stable LiTi*_x_*Mn_2−*x*_O_4_ framework, which complies well with the XRD results. In addition, there is a new phase similar to TiMn_2_O_4_ formed at the spinel LiMn_2_O_4_ (001) surface.

It is known that the surface energy is gradually reduced in the sequence of {001}, {110} and {111} [48], thus the presence of the TiMn_2_O_4_-like spinel phase on the (001) surface may be related to the surface energy difference. Thus, the {001} plane is in accordance with the most unstable surfaces, favoring the Ti cations shift. Moreover, the Li-terminated LiMn_2_O_4_ {001} surfaces are also very unstable due to the increased dangling bonds and lower bonding energy with the oxygen anions [49]. Therefore, a small amount of TM cations can exchange the position with Li^+^, resulting in the formation of reconstruction layers in these regions. This reconstruction layers (TiMn_2_O_4_-like) are able to produce a more stable cathode/electrolyte interfacial layer due to the stronger Ti-O bond, promoting the stability of cathode materials [21].

To further unveil the change in surface chemical states around different crystal planes, the pristine LiMn_2_O_4_ and LiTi_0_._5_Mn_1_._5_O_4_ samples are characterized using EELS, and the results are shown in Figure 7a–c. Figure 7a shows the Ti-L_2,3_, O-K, and Mn-L_2,3_ energy-loss near-edge fine structure (ELNES) around the (111) facet surface for the LiMn_2_O_4_, and the (111), (110), and (001) facet surfaces for the LiTi_0_._5_Mn_1_._5_O_4_ after background subtraction and normalization. In LiTi_0_._5_Mn_1_._5_O_4_, the pre-peak intensity of the O-K edge in LiTi_0_._5_Mn_1_._5_O_4_ is slightly less than that in LiMn_2_O_4_, which is correlated with a slight decrease in Mn valence [50]. Moreover, the O-K spectrum in LiMn_2_O_4_ shows a sharp peak followed by a shoulder structure, while two peaks at 532.4 and 534.2 eV in LiTi_0_._5_Mn_1_._5_O_4_ are observed, which can be assigned to the transition to the 3d bands of tetravalent Ti. Four peaks in the Ti-L_2,3_ ELNES for LiTi_0_._5_Mn_1_._5_O_4_ at different facets are also shown in Figure 7a, a fingerprint of Ti^4+^, which is also consistent with the XPS measurement. By using the pristine LiMn_2_O_4_ as a reference to extract the k-factors (Appendix A), the Mn/O ratios (R_Mn/O_) at different surface planes were quantified (Figure 7b), in which R_Mn/O_(110) > R_Mn/O_(111) > R_Mn/O_(001) planes. In the (001) plane, R_Mn/O_(001) is approximately 0.39, indicating that Ti^4+^ is enriched in the (001) plane. Furthermore, the relationship between the Mn (L_3_/L_2_) intensity ratio and the Mn valence state at different facets is investigated (Figure 7c). A higher L_3_/L_2_ value results in a decreased Mn valence state [51,52,53,54], originating from the Ti doping effect as claimed in Figure 7a.

## 4. Conclusions

In conclusion, Ti-doped truncated octahedron LiMn_2_O_4_ samples were synthesized by a facile hydrothermal treatment and calcination process. Cs-STEM and chemical analysis techniques were carried out to reveal the underlying mechanism of Ti doping on the structure evolution and the stability enhancement of LiMn_2_O_4_ samples with different contents of Ti doping. It is found that Ti doping is beneficial to forming truncated octahedron LiTi*_x_*Mn_2−*x*_O_4_. Among the samples, LiTi_0_._5_Mn_1_._5_O_4_ samples exhibit the most obvious truncated octahedron structure. After Ti ion doping, the percentage of Mn^4+^ in LiTi_0_._5_Mn_1_._5_O_4_ reduced, suggesting the successful replacement of Mn^4+^ with Ti^4+^. Based on detailed surface structural analysis of the {111}, {110}, and {001} planes of LiTi_0_._5_Mn_1_._5_O_4_ at the atomic scale, it is found that there is a more stable spinel LiTi*_x_*Mn_2−*x*_O_4_ framework formed in bulk, as well as at the (111) and (110) planes. In addition, a TiMn_2_O_4_-like structure at the {001} surface is observed and thoroughly analyzed by Cs-STEM combined with EELS techniques. The new TiMn_2_O_4_-like structure can reduce the surface energy of (100) planes and accelerate the growth rate of (100) planes, therefore enhancing the stability of the spinel LiMn_2_O_4_. According to the EELS analysis, the appearance of the TiMn_2_O_4_-like phase can be associated with the enrichment of Ti^4+^. Our findings demonstrate the critical role of the Ti ion doping in the surface chemical and structural evolution of LiMn_2_O_4_, which provides a facile method for high-stability cathode materials design and growth.

## Figures and Tables

**Figure 1 nanomaterials-11-00508-f001:**
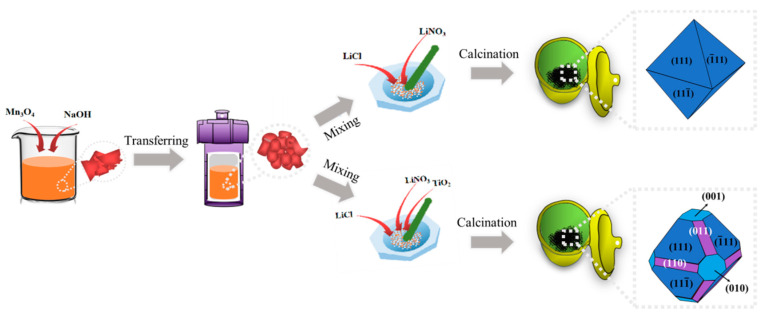
Schematic diagram of the preparation process of the LiTi*_x_*Mn_2−*x*_O_4_ samples.

**Figure 2 nanomaterials-11-00508-f002:**
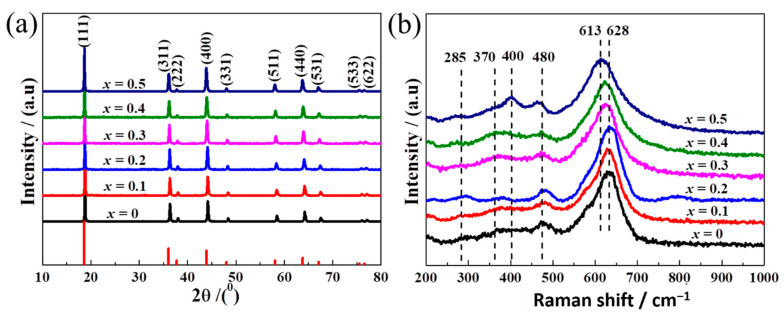
XRD patterns **(a)** and Raman spectra **(b)** of the LiTi*_x_*Mn_2−*x*_O_4_ (*x* = 0, 0.1, 0.2, 0.3, 0.4, 0.5) samples.

**Figure 3 nanomaterials-11-00508-f003:**
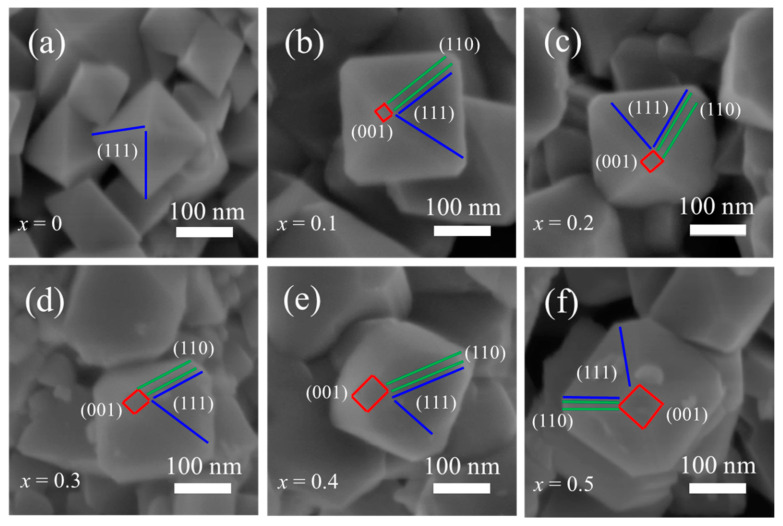
SEM images of LiTi*_x_*Mn_2−*x*_O_4_ at different concentrations (*x* = 0, 0.1, 0.2, 0.3, 0.4, 0.5) (**a–f**).

**Figure 4 nanomaterials-11-00508-f004:**
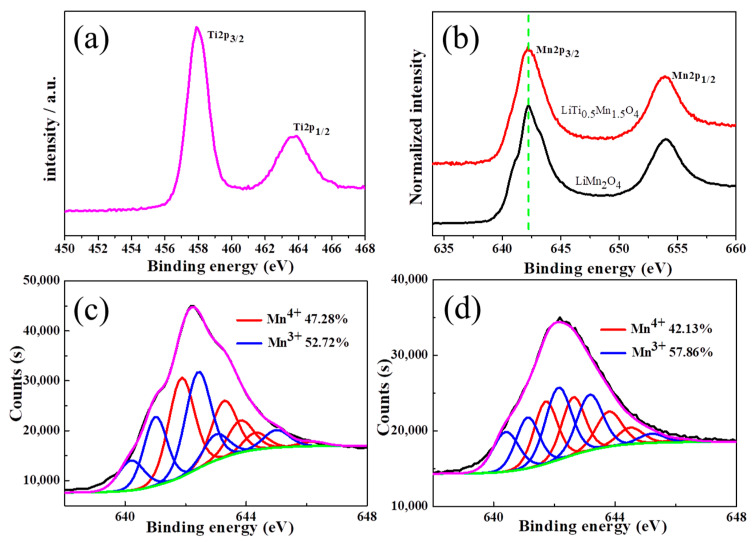
Ti 2p XPS spectra of LiTi_0_._5_Mn_1_._5_O_4_ (**a**). Mn 2p XPS spectra (**b**) of LiMn_2_O_4_ and LiTi_0_._5_Mn_1_._5_O_4_. Fitted spectra of LiMn_2_O_4_ (**c**) and LiTi_0_._5_Mn_1_._5_O_4_ (**d**).

**Figure 5 nanomaterials-11-00508-f005:**
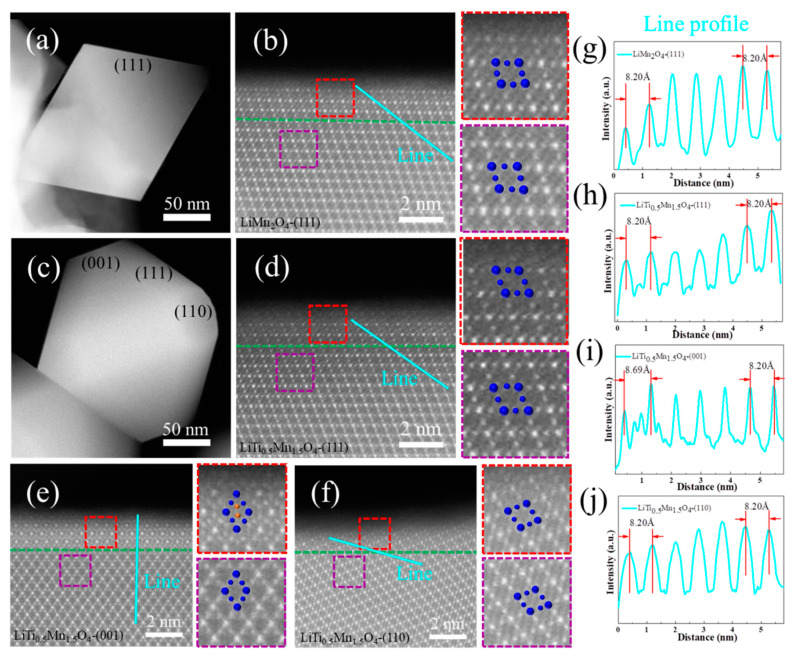
Low-magnification (a) and high-resolution (b) high-angle annular dark-field (HAADF) images of the LiMn_2_O_4_ viewed from the [110] crystallographic direction in (111) planes. Low-magnification (c) and high-resolution HAADF images of the LiTi_0_._5_Mn_1_._5_O_4_ particles viewed from the [110] crystallographic direction in the (111) plane (d), (001) plane (e), and (110) plane (f). Magnified views of selected regions are shown in the right panels, where the contrast corresponding to the Mn columns at 16d and 8a sites are indicated by blue and orange spheres, respectively. The boundary between the bulk and the surface regions is marked by the green dashed line. Line profiles (g–j) correspond to the sky blue lines in panel (b,d–f), respectively.

**Figure 6 nanomaterials-11-00508-f006:**
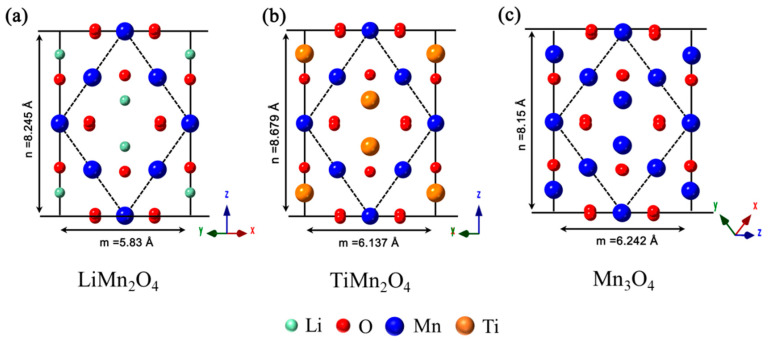
Crystal structure of the LiMn_2_O_4_ (a), TiMn_2_O_4_ (b), and Mn_3_O_4_ (c) viewed along the [110] direction.

**Figure 7 nanomaterials-11-00508-f007:**
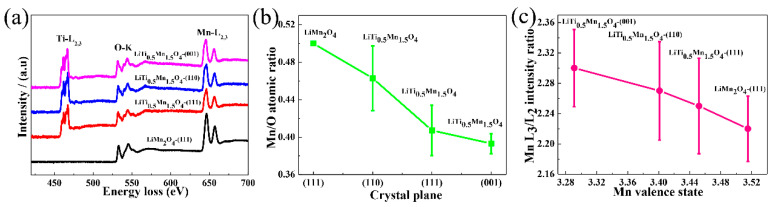
ELNES spectra of Ti-L_2,3_, O-K, and Mn-L_2,3_ from the surface of the (111) facet of the LiMn_2_O_4_, and (111), (110), and (001) facets of the LiTi_0_._5_Mn_1_._5_O_4_ (**a**). Mn/O atomic ratio of LiTi_0_._5_Mn_1_._5_O_5_ (**b**). Pristine LiMn_2_O_4_ was used as a reference to extract the k factors. Dependence of the Mn (L_3_/L_2_) intensity ratio vs. the Mn valence state in the (111) facet of the LiMn_2_O_4_, and (111), (110), and (001) facets of the LiTi_0_._5_Mn_1_._5_O_4_ (**c**).

## Data Availability

The data presented in this study are openly available in [repository name e.g., FigShare] at [**doi**], reference number [reference number].

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
