# Peer review of "Atomic Insights into Ti Doping on the Stability Enhancement of Truncated Octahedron LiMn2O4 Nanoparticles"

_nanomaterials, 2021, doi:10.3390/nano11020508_

Round 1

Reviewer 1 Report

This article was written on the subject of Ti particle doping for LiMn2O4 nanoparticles. In addition, a basic
hydrothermal treatment and calcination procedure has been used to synthesize Ti-doped truncated octahedron
LiTixMn2-xO4 nanocomposites. Different treatment processes have been carried out and proposed to shape the
above-mentioned nano-particles, but some of the information is misleading. The article is in good shape, but on
some of the sections before publication, the article requires a more stable and transparent approach. Please , find
some of the corrections and recommendations.
1. It is not clear from the manuscripts why especially the doping of the Ti has been chosen. Please briefly
explain.
2. Introduction lacks literature on Ti properties and needs more information.
3. Hydrothermal treatment and calcination process needs more explanation because potential readers may
need more information on the process to further clarify the treatment process.
4. “The combination of LiTixMn2-xO4 frameworks in bulk and the 20
TiMn2O4-like structure at the surface may enhance the stability of the spinel LiMn2O4”. In the above
sentence, according to authors, stability might get affected in positive terms. Please mention some
practical applications where this has been proved.
5. Please rename the section “Results” to “Results and discussions”.
6. Please enhance the quality of Figure 7.

Reviewer 2 Report

In this manuscript, the preparation and characterisation of the studied materials are described at high level. The microscopic data are especially impressive. It would be advantageous to present some electrochemical data but I guess that it will be a subject of another paper.

Furthermore, the paper is clearly written and well organized. The materials are well characterized, and I think that the paper can be accepted as it is.

Reviewer 3 Report

Ti-doping of spinel LiMn2O4 and the effect of this doping on the transformation of the spinel octahedrons into truncated LiTixMn2-xO4 octahedrons have been studied experimentally. The authors managed to find the most promising composition of the doped material and explain the mechanism responsible for the structural changes.

Being not an expert in the field, I cannot judge the potential practical importance of this study. From the academic point of view, the results are interesting. The paper is well written and easy to read. I do not have any question about the quality of the manuscript.
